# OpenReview forum: "SOS! Soft Prompt Attack Against Open-Source Large Language Models"
_ICLR.cc/2026/Conference — ICLR 2026 Conference Withdrawn Submission_

### Official Review · Reviewer_XSnF · 2025-10-17

**Soundness:** 2
**Presentation:** 3
**Contribution:** 2
**Rating:** 2
**Confidence:** 4

**Summary:**

This paper proposes SOS attack paradigm, where the attacker first optimizes soft prompts and replaces the token embeddings with the soft tokens. This paradigm enables multiple attack objectives, backdoor, adversarial attack, and prompt-stealing attacks. An exploration of dual potential to safeguard copyrighted content is conducted.

**Strengths:**

- This paper proposes an SOS attack, which targets the token embedding layer.
- The methodology is rational and features requiring no clean data.

**Weaknesses:**

- The models under evaluation are too old. Evaluating on more recent models is in demand.
- For the threat model, it is a classical backdoor scenario. But why does the attacker prefer modifying the *soft prompt tokens* rather than directly fine-tune the whole model?
- The references for backdoor attacks in the context of generative LLMs are out of date, e.g., referring to a survey [1][2].

Reference
- [1] [A Survey of Recent Backdoor Attacks and Defenses in Large Language Models](https://arxiv.org/abs/2406.06852)
- [2] [A Survey on Backdoor Threats in Large Language Models (LLMs): Attacks, Defenses, and Evaluations](https://arxiv.org/abs/2502.05224)

**Questions:**

See weaknesses.

---

### Official Review · Reviewer_ARQm · 2025-10-23

**Soundness:** 2
**Presentation:** 2
**Contribution:** 2
**Rating:** 4
**Confidence:** 4

**Summary:**

This paper introduces SOS, a novel and efficient framework that targets the token embedding layer of large language models (LLMs). Instead of changing model weights or requiring clean data, SOS optimizes a small set of adversarial embeddings and maps them to rare trigger tokens to enable a range of attacks such as adaptive backdoor, jailbreak, and prompt stealing while keeping computation low and preserving model utility. Experiments on various LLMs show near perfect attack success rates and strong transferability. The authors also demonstrate a protective use case for content and model copyright protection by publishing copyright tokens that cause protected models to ignore marked content.

**Strengths:**

- SOS introduces a lightweight, embedding-based attack approach that avoids fine-tuning model weights while maintaining model utility.
- SOS requires fewer samples and minimal computational resources to reach high attack success rates.
- SOS supports multiple attack types, including backdoor, jailbreak, and prompt stealing within a unified framework.

**Weaknesses:**

- The paper claims to be “first” to systematically target token-embedding layers, but prior works has explored poisoned embeddings threats. The paper does not clearly delineate how SOS meaningfully departs from or improves over these [1, 2].
- Although the method claims stealth (model utility preserved when triggers absent), the paper provides only limited defense evaluation (ONION) and no systematic study of detectability by embedding-level integrity checks, model fingerprinting defenses, or distributional-monitoring approaches. This weakens security conclusions.
- A simple defender strategy (comparing or hashing token-embedding layers against a trusted clean release) could detect the injected embeddings but is not evaluated.

[1] Huang, Y., Zhuo, T. Y., Xu, Q., Hu, H., Yuan, X., & Chen, C. (2023, April). Training-free lexical backdoor attacks on language models. In Proceedings of the ACM Web Conference 2023 (pp. 2198-2208).
[2] Guo, D., Hu, M., Guan, Z., Guo, J., Hartvigsen, T., & Li, S. (2024). Backdoor in Seconds: Unlocking Vulnerabilities in Large Pre-trained Models via Model Editing. CoRR.

**Questions:**

1. Can the attack be detected by comparing embedding norms, cosine similarities, or spectral distributions between clean and attacked models?
2. How does SOS perform when applied to models with different tokenizer vocabularies (e.g., BPE vs. SentencePiece)?
3. What is the effect of adding multiple trigger tokens? Does it scale linearly in attack success rate or interact nonlinearly with the model’s contextual embedding behavior?
4. Does SOS generalize to multimodal or instruction-finetuned models where embeddings are aligned with vision or code tokens?
5. How sensitive is the attack to the choice of adversarial loss (e.g., NLL vs. contrastive objectives) or the size of the crafted dataset?

**Details Of Ethics Concerns:**

The SOS method describes a practical way to tamper with token embedding matrices of open source LLMs so that models can be covertly altered to produce harmful outputs, reveal confidential prompts, or spread misinformation. Withholding the actual embeddings helps, but the method is simple enough to be reproduced from the description, creating clear misuse risk. The manuscript also frames embedding manipulation as a protective technique for copyright, which may normalize the approach without setting guardrails. The authors do not provide a rigorous risk assessment and do not evaluate concrete detection or mitigation strategies. I recommend the authors add detailed mitigations and detection evaluations before public release.

---

### Official Review · Reviewer_Sc9j · 2025-10-27

**Soundness:** 2
**Presentation:** 2
**Contribution:** 2
**Rating:** 4
**Confidence:** 4

**Summary:**

This paper proposes an attack method targeting the token-embedding layer of LLMs. By employing soft prompt tuning, it learns a series of adversarial embeddings that enable backdoor, jailbreak, and prompt-stealing attacks. Experiments are conducted to demonstrate the effectiveness of the proposed approach.

**Strengths:**

1. The paper presents a novel perspective by attacking LLMs from the token-embedding layer, which is relatively unexplored.

2. The use of soft prompt tuning offers high computational efficiency and helps maintain model performance.

**Weaknesses:**

1. Insufficient literature review: The paper lacks a thorough survey of existing attack methods. For example, for backdoor attacks, it does not discuss recent work such as Uncertainty is Fragile [1] and Backdoor Threats to LLM-based Agents [2], etc. ; similarly, for jailbreak attacks, it overlooks recent methods such as FlipAttack [3] and X-Teaming [4], etc. .

2. Limited novelty: The method essentially fine-tunes soft prompts on specific adversarial datasets without providing deeper theoretical insights. However, it can be acknowledged that the paper is among the first to systematically apply such techniques to the embedding layer across multiple attack scenarios.

3. Inadequate experimental comparison: The baselines are relatively weak, focusing mainly on comparisons with the original model or earlier approaches such as GCG and AutoDAN. The paper should include comparisons with more recent SOTA methods (e.g., [4] X-Teaming for jailbreak) to strengthen its empirical claims.

References:
 [1] Zeng et al., Uncertainty is Fragile: Manipulating Uncertainty in Large Language Models, arXiv:2407.11282, 2024.
 [2] Yang et al., Watch Out for Your Agents! Investigating Backdoor Threats to LLM-based Agents, NeurIPS 2024.
 [3] Liu et al., FlipAttack: Jailbreak LLMs via Flipping, arXiv:2410.02832, 2024.
 [4] Rahman et al., X-Teaming: Multi-turn Jailbreaks and Defenses with Adaptive Multi-Agents, arXiv:2504.13203, 2025.

**Questions:**

See weakness part

---

### Official Review · Reviewer_M1QX · 2025-10-29

**Soundness:** 3
**Presentation:** 1
**Contribution:** 2
**Rating:** 2
**Confidence:** 3

**Summary:**

This paper proposes a soft-prompt–based attack framework that targets the embedding layer of open-source LLMs. By optimizing a small number of adversarial embeddings and binding them to rare trigger tokens, the method enables backdoor, jailbreak, and prompt-stealing attacks with very low data and compute requirements. The authors also claim that this framework can be repurposed for copyright protection and model fingerprinting, illustrating both risks and opportunities in LLM embedding vulnerabilities.

**Strengths:**

1. The conceptual framework is versatile and supports multiple adversarial goals (backdoor, jailbreak, and prompt stealing), as well as proposed benign use cases.
2. The proposed method is efficient: it requires only ~10 training samples and does not update internal model weights.
3. The reported empirical performances (attack success rates, prompt reconstruction metrics, etc.) of different attacks are impressive.

**Weaknesses:**

1. Limited novelty: the paper’s central technique (optimizing continuous embedding/soft-prompt vectors and assigning them to trigger tokens) closely follows recent embedding-space attack literature (e.g., https://arxiv.org/pdf/2402.09063, Schwinn et al., 2024, which the authors also cited). For example, the statement in the abstract that token-embedding vulnerabilities are “underexplored” overstates novelty. Although the authors acknowledge this concern in section 8, they should more clearly delineate what is new and explicitly compare this work to the most closely related papers with more technical details. Additionally, all references in this paper are from 2024 or earlier, so the literature review should be updated to include more recent developments since 2025.
2. Writing and structure need improvement: for example, section 3 is not very informative and adds little value as a standalone section, while important methodology details in section 4.2 remain relatively high-level, which might make readers unfamiliar with the attacks hard to understand.
3. The evaluated models are somewhat outdated: the experiments do not include recent open-source families (e.g., newer Llama models, Olmo, and Gemma), which reduces confidence in SOS’s generalizability.

**Questions:**

1. Since it’s mentioned in the Appendix that the choice of tokens could affect attack performance, did you try multiple trigger tokens beyond the one you used in your main experiments?
2. Could the authors provide mechanistic explanations or plausible hypotheses for why such limited resources used to fine-tune the adversarial embedding can still be highly effective?

---

### Official Review · Reviewer_311v · 2025-11-01

**Soundness:** 2
**Presentation:** 3
**Contribution:** 2
**Rating:** 4
**Confidence:** 4

**Summary:**

This paper studies soft prompt attacks in a broad context, applying the established technique of optimizing soft prompts [1, 2] to target several undesirable behaviours such as backdoor jailbreaks, and prompt stealing. Beyond a fairly comprehensive suite of experiments across these different settings, the key contribution of this paper is to operationalize the learned soft prompts through a simple idea of replacing the embeddings of underused or new tokens in the model vocabulary with these learned embeddings, thereby inserting a set of backdoor “trigger tokens” that someone can use to trigger certain behaviour.

The main threat model identified is the backdoor setting, where an entity trains a useful model, then embeds these triggers into the model, and that entity can manipulate any system built using their compromised LLM. To the best of my knowledge, it would be very difficult for someone to check for the existence of backdoors in an LLM, making it relevant threat model to study, though it does require an intentionally adversarial model provider to deliberately implant these triggers.

Embedding these backdoor trigger tokens in the model’s embedding layer is a simple and (to the best of my knowledge) novel way to operationalize these soft prompts. Experimentally, we would expect the performance of these soft prompts to be equivalent to a soft prompt suffix attack [2], as the only difference is that the embeddings are copied to a real discrete token’s corresponding embedding. ASR metrics are therefore typically high especially in specific/targetted settings, in line with previous results.

While I think the overall approach and idea is good, I unfortunately have some concerns with the framing of the paper and the experimental choices made. My main concern is that I believe the threat model makes the most sense in an agentic setting, or a setting in which the LLM can make decisions/can execute tasks (for instance, embedding a trigger in an LLM used in agentic settings to issue a refund, make certain API calls that could reveal private information, etc.). While in theory the proposed technique *should* work in this setting, there are no experimental results to support this. The concern is how well such prompts can generalize to more complex tasks that may be required of an agent, as the expressivity of the a few extra soft tokens may be insufficient [3]. Also note that in the safer model (Llama 2 vs Alpaca/Mistral), the ASR of the jailbreak results is weaker, suggesting that my aforementioned concern may indeed be an issue (since a “universal” trigger likely requires more expressivity to be able to adapt to general queries).

I also have concerns with the results of some of the evaluations. In their misinformation example, they argued that the trigger token can be used to to trigger misinformation, but in their example what seems to be happening is the trigger token generates the sequence it was optimized to generate (i.e. the “targeted backdoor” setting), while the rest of the content doesn’t seem to make use of this text. In order for it to be adaptive, there should be evidence of the subsequent generation leveraging this misinformation, or more ideally, adaptively produces misinformation in line with some ***higher level goal*** that the adversarial triggers were trained to produce, rather than a single specific concept. Finally, the results are primarily on older models; there have been substantial improvements to model safety training since then which would be important to evaluate for.

Because of the above concerns, I currently recommend rejection. I would encourage the authors to explore this idea in more interactive settings where such triggers could be better utilized, eg. triggering tool calls.

[1] Lester, Brian, Rami Al-Rfou, and Noah Constant. "*The power of scale for parameter-efficient prompt tuning.*" (2021)

[2] Schwinn, Leo, et al. "*Soft prompt threats: Attacking safety alignment and unlearning in open-source llms through the embedding space*." (2024)

[3] Li, Xiang Lisa, and Percy Liang. "*Prefix-tuning: Optimizing continuous prompts for generation.*" (2021).

**Strengths:**

- Good core idea of operationalizing soft prompts: the approach of replacing embeddings of underused/new tokens with learned soft prompt embeddings to create backdoor "trigger tokens" appears to be a simple and novel contribution
- Comprehensive experimental scope: the work applies soft prompt optimization across multiple attack scenarios (Target/adaptive backdoor, jailbreaks, prompt stealing), as well as two benevolant use cases: content and model copyright protection.
- Relevant threat model: backdoor scenario represents a realistic security concern
- Strong baseline performance: ASR metrics align with previous soft prompt suffix attacks [2]

[2] Schwinn, Leo, et al. "*Soft prompt threats: Attacking safety alignment and unlearning in open-source llms through the embedding space*." (2024)

**Weaknesses:**

**Misalignment between threat model and experiments**
In my opinion, the most compelling use case of the *novel* aspects of this work (a simple technique to operationalize soft prompts to manipulate LLMs) is to manipulate agentic/decision-making LLMs, and this lacks experimental validation. No results demonstrate effectiveness in settings where agents execute tasks or make consequential decisions.
  - Note that previous works [2] have already studied soft prompts in the context of jailbreaking (without the trigger token mapping contribution), so the jailbreak/trigger *performance* results are not novel.

**Unclear generalization to complex tasks**
I’m concerned that the expressivity of soft tokens may be insufficient for more complex behaviour; no evidence provided to address this, and weaker ASR on safer models (Llama 2) suggests the approach may struggle with more robust models.

**Outdated experimental setting**
The approach is evaluated on older models, and modern safety training improvements not assessed.

[2] Schwinn, Leo, et al. "*Soft prompt threats: Attacking safety alignment and unlearning in open-source llms through the embedding space*." (2024)

**Questions:**

- What are your thoughts of applying this to a more interactive setting, especially settings like tool use or multi-step tasks? I think being able to trigger certain tool calls (especially without knowing the tool schema) would strengthen the results of this paper.
- Can you provide clearer examples or metrics showing that the model *adaptively uses* the triggered misinformation rather than just reproducing it? The current evidence doesn't convincingly demonstrate the threat; jailbreaking in some sense is a better example for this type of universal behaviour, but is not the same thing.
- Why did your results suffer so much with Llama 2 over the other models? Have you tried more recent models?

---

### Note · Authors · 2025-11-17

I have read and agree with the venue's withdrawal policy on behalf of myself and my co-authors.